# SOME NEURAL NETWORKS INHERENTLY PRESERVE SUBSPACE CLUSTERING STRUCTURE

**Karan Vikyath Veeranna Rupashree**[*]**, & Siddharth Baskar**[*]
Department of Electrical and Computer Engineering
University of Wisconsin-Madison
Madison, Wisconsin, USA
{veerannarupa,sbaskar2}@wisc.edu

**Daniel L. Pimentel-Alarcón**
Department of Biostatistics and Medical Informatics
University of Wisconsin-Madison
Madison, Wisconsin, United States
pimentelalar@wisc.edu

## ABSTRACT

It has long been conjectured and empirically observed that neural networks tend to preserve clustering structure. This paper formalizes this conjecture. Specifically, we establish precise conditions for cluster structure preservation and derive bounds to quantify its extent. Through this analysis we are able to show that certain neural networks are learning parameters that preserve the clustering structure of the original data in their embeddings, without the need to impose mechanisms to promote this behavior. Extensive numerical analysis and experiments validate our results. Our findings offer deeper insight into neural network behavior, explaining why certain data types (such as images, audio, and text) benefit more from deep learning. Beyond theory, our findings guide better initialization, feature encoding, and regularization strategies.

## 1 INTRODUCTION

At this point neural networks (NNs) need no introduction. They are the backbone of modern artificial intelligence (AI), revolutionizing industries and reshaping the modern world. Their applications span diverse fields: in healthcare, NNs enhance medical imaging, predictive analytics (Almarzouqi et al., 2022; Khang et al., 2024), and drug discovery, accelerating the identification of new therapeutics (Wong et al., 2024). In finance, they play a crucial role in fraud detection and algorithmic trading (Wang et al., 2021; Luo et al., 2024). Beyond these domains, NNs are driving scientific innovation, predicting protein structures (Zhou et al., 2024) and improving climate modeling (Ghimire et al., 2022). Convolutional neural networks (CNNs) revolutionized image processing (Chauhan et al., 2018), recurrent neural networks (RNNs) reinvented sequential data processing (Al-Selwi et al., 2024), and attention mechanisms (Vaswani, 2017) transformed natural language processing, enabling advancements in machine translation (Wang et al., 2022), sentiment analysis (Sachin et al., 2020), and conversational AI (Saka et al., 2023). Meanwhile, generative adversarial networks (GANs) and diffusion models are redefining art and content creation (Goodfellow et al., 2020; Croitoru et al., 2023).

Despite the widespread use of neural networks, their inner workings, particularly their efficiency in clustering and predictive tasks, remain perplexing. Various studies have highlighted these challenges. For instance, it is unclear how neural networks can achieve high performance even when trained on randomized labels (Song et al., 2022), or how phenomena like double descent allows model performance to improve with complexity beyond the point of overfitting (Belkin et al., 2019), or how weight initialization affects stochastic gradient descent (Narkhede et al., 2022; Bishop & Bishop, 2023).

Among these challenges, one major obstacle in deep learning theory is a detailed understanding of the complexities introduced by high-dimensional and non-linear transformations, which obscure how these models arrive at their decisions (Doshi-Velez & Kim, 2017; Lipton, 2018). These enigmatic challenges are fundamentally rooted in activation functions like the sigmoid, the hyperbolic tangent, or the Rectified Linear Unit (ReLU) (Hara et al., 2015). These functions introduce non-linearity to NNs,

---

[*]Equal Contribution

required for hierarchical feature learning and to represent highly complex predictive functions. The downside is that activation functions also impact loss functions, affecting optimization dynamics and complicating theoretical analysis. Recent studies have explored the expressivity and approximation properties of ReLUs to provide new insights on their effect on neural network performance (Kou et al., 2024). However, a clear understanding of the fundamental role of activation functions and their effect on the behavior of NNs remains elusive.

**This paper** describes specific conditions under which certain activation functions preserve subspace clustering structure. More precisely, we characterize conditions to guarantee that if the input to a layer with a valid activation function has a subspace cluster structure, the output will retain that same cluster structure in the output's embedding. One example of such activation function is the *rectified linear unit* (ReLU). This behavior has been long conjectured and empirically observed in many datasets (Papyan, 2020; Arora et al., 2018). Besides formalizing this conjecture, our analysis gives a deeper understanding of the underlying reasons for this behavior, providing insights into initialization parameters that promote cluster preservation. Our analysis also explains why NNs perform better for certain types of data, such as imagery, audio (Nanni et al., 2021), text data for natural language processing (Min et al., 2023), bioinformatics data (Karim et al., 2021), and financial and anomaly detection (Zamanzadeh Darban et al., 2024). It turns out that these types of data inherently preserve subspace clustering structure under certain transformations.

The proof brings together ideas from statistical learning, principal component analysis (PCA), subspace clustering (SC), and perturbation theory, and it is divided in three main parts: (i) first we show that the closed-form solution to a subspace clustering model (of which Euclidean clustering is a special case) can be accurately inferred from noisy data. (ii) Then we show that this solution is invariant to arbitrary linear transformations. (iii) Lastly, we show that under certain conditions on the network parameters and the activation function, such solution is robust to the corresponding transformation. We establish precise conditions for cluster structure preservation and derive bounds to quantify its extent, leveraging the Davis-Kahan $\sin(\boldsymbol{\Theta})$ theorem. Numerical results confirm these bounds, and experiments further validate that ReLUs and several other related activation functions inherently preserve clustering structure. While our theoretical guarantees apply to a single layer, they extend directly to multiple layers, as supported by our empirical findings.

**Ultimately**, this paper brings to the table new insights and a deeper understanding into the inner workings of neural networks and how they learn. Specifically, the main takeaway from our findings is that neural networks that use ReLUs and similar activation functions seem to be learning to cluster in closed form.

## 2 RELATED WORK

The theoretical understanding of neural networks, due to their high-dimensional and non-linear structures, still remain an ongoing challenge. While the deep learning models achieve high performance, their interpretability and underlying mechanism are still not fully understood. Methods like feature attribution (Lundberg, 2017; Molnar, 2020) and mechanistic interpretability (Bereska & Gavves, 2024) have attempted to explain how neural networks process information. However, these methods often fail to capture the full structural properties.

The ability of deep networks to achieve high performance even when trained on randomized labels (Asnicar et al., 2024) further complicates the understanding of learned representations. One contributing factor is overparameterization, where large networks manage to preserve feature structures despite having more parameters than necessary (Elhage et al., 2021). Additionally, the implicit biases introduced by optimizers such as stochastic gradient descent (SGD) have been linked to the preservation of clustering structures, suggesting that optimization dynamics play a crucial role in shaping learned representations (Soudry et al., 2018).

Double descent is a phenomenon (Belkin et al., 2019; Nakkiran et al., 2021), in which performance initially deteriorates as the complexity of the model increases but improves again after exceeding the overfit threshold. Recent studies indicate that double descent may be linked to how activation functions structure the feature space, hence preserving the cluster boundaries (Advani et al., 2020; Zhang et al., 2021).

Additionally, network initialization and optimization strategies affect the extent to which clustering structures are preserved. Techniques such as Xavier and Kaiming initialization (Pan et al., 2022) help maintain stable gradients, indirectly influencing feature separability. Recent work on flat minima and generalization (Ding et al., 2024) has shown that flatter loss landscapes are associated with better-preserved clustering structures, a property that may be influenced by activation functions like ReLU. Another work related to activation functions highlights the importance of non-linearities in preserving structures such as clusters. This also states that ReLU preserves topological features in latent spaces, enhancing the clustering of data points in learned representations (Xu, 2015).

## 3    MAIN RESULTS

Consider a data matrix $\mathbf{X}^\star \in \mathbb{R}^{m \times n}$ with columns given by

$$\mathbf{x}_i^\star = \sum_{k=1}^{K} \mathbb{1}_{\{i \in \mathbf{\Omega}_k\}} \mathbf{U}_k \, \mathbf{v}_i,$$

where $\mathbb{1}$ denotes the indicator function, $\{\mathbf{\Omega}_k\}_{k=1}^{K}$ is a partition of $\{1, \ldots, n\}$ indicating the clustering of the columns among K subspaces with bases $\mathbf{U}_k \in \mathbb{R}^{m \times r_k}$, and $\mathbf{v}_i \in \mathbb{R}^{r_k}$ is the vector of coefficients of $\mathbf{x}_i^\star$ with respect to the basis $\mathbf{U}_k$. These data have a subspace cluster structure where each sample lies in one of K low-dimensional subspaces. This model is often known as a *union of subspaces* (UoS) model (Lipor & Balzano, 2017), generalized PCA (Vidal et al., 2005) or subspace clustering (Elhamifar & Vidal, 2013). Suppose our observed data is

$$\mathbf{X} = \mathbf{X}^\star + \mathbf{Z}, \tag{1}$$

where $\mathbf{Z} \in \mathbb{R}^{m \times n}$ can be interpreted as a noise matrix, so that the columns in $\mathbf{X}$ lie *near* the UoS, rather than exactly on it. Notice that Euclidean clustering is the special case of (1) with 1-dimensional subspaces and constant coefficients (i.e., $\mathbf{U}_k \in \mathbb{R}^m$, often denoted as $\boldsymbol{\mu}_k$, and $\mathbf{v}_i = 1$ for every i). Similarly, orthogonal nonnegative matrix completion (ONMF) (Ding et al., 2006), also used for its clustering capabilities (Pompili et al., 2014), is the special case of (1) with K = n 1-dimensional orthogonal subspaces and $\mathbf{U}_k \geq \mathbf{0}$ for every k and $\mathbf{v}_i \geq 0$ for every i.

Our main result, summarized in Theorem 3.1 below, specifies sufficient conditions under which certain layers preserve the subspace cluster structure described above. More formally, let $\sigma(\cdot) := \max(\mathbf{0}, \cdot)$ denote the layer's activation function, and let $\mathbf{W}$ denote the parameters of the layer, so that $\mathbf{Y} = \sigma(\mathbf{W}\mathbf{X})$ is the output of the layer when $\mathbf{X}$ is fed (we are omitting the bias term $\mathbf{b}$, which can be incorporated as a column of $\mathbf{W}$ by adding a constant row of ones in $\mathbf{X}$). Given a matrix, let $\mathbf{P}(\cdot)$ denote the projection operator onto its principal row-space of dimension $r := \sum_{k=1}^{K} r_k$. We will show that $\mathbf{P}$ encodes the subspace cluster structure above. The next theorem shows that under reasonable conditions, the projection matrices $\mathbf{P}(\mathbf{X}^\star)$ and $\mathbf{P}(\mathbf{Y})$ are close enough to one another and hence they encode the same clustering structure.

---

**Theorem 3.1.** *Let $\delta(\mathbf{X}^\star, \mathbf{X})$ denote the gap between the $r^{\text{th}}$ singular value of $\mathbf{X}^\star$ and the $(r+1)^{\text{th}}$ singular value of $\mathbf{X}$, and similarly for $\delta(\mathbf{W}\mathbf{X}^\star, \mathbf{W}\mathbf{X})$ and $\delta(\mathbf{Y}, \mathbf{W}\mathbf{X})$. Suppose the smallest of these quantities, $\delta$, satisfies:*

$$\delta > \frac{\sqrt{2^7 r}}{\epsilon} \max(\|\mathbf{Z}\|, \|\mathbf{W}\mathbf{Z}\|, \|\mathbf{Y} - \mathbf{W}\mathbf{X}\|) =: \eta. \tag{2}$$

*Then*

$$\|\mathbf{P}(\mathbf{X}^\star) - \mathbf{P}(\mathbf{Y})\|_\infty < \epsilon/2.$$

*Furthermore, $\mathbf{x}_i^\star, \mathbf{x}_j^\star \in \text{span}(\mathbf{U}_k)$ if and only if the $(i, j)^{\text{th}}$ entry of $\mathbf{P}(\mathbf{Y})$ is larger than $\epsilon/2$.*

---

The proof of Theorem 3.1 is the next Section, and it follows by the Davis-Kahan $\sin(\mathbf{\Theta})$ Theorem (Davis & Kahan, 1970; Stewart & Sun, 1990). Intuitively, $\delta$ can be interpreted as the similarity between the principal subspaces of its arguments. Under this light, Theorem 3.1 requires (i) that $\mathbf{X}$ is

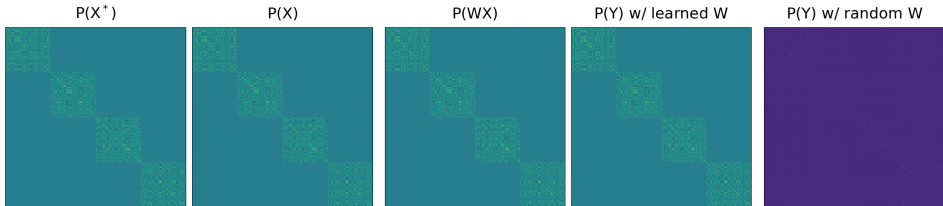

Figure 1: Projection matrices revealing the clustering structure of the data, which is preserved through every step of a ReLU transformation. Lemma 4.1 shows that the projections of $\mathbf{X}^\star$ and $\mathbf{X}$ are close. Lemma 4.2 shows that the projections of $\mathbf{X}$ and $\mathbf{WX}$ are close. Lemma 4.3 shows that the projections of $\mathbf{WX}$ and $\mathbf{Y}$ are close when $\mathbf{W}$ satisfies certain conditions that the network encourages through the learning process. In contrast, random $\mathbf{W}$'s destroy the clustering structure.

close enough to $\mathbf{X}^\star$, which depends on the noise level in $\mathbf{Z}$, (ii) that $\mathbf{X}$ and $\mathbf{X}^\star$ behave similarly under the linear transformation $\mathbf{W}$, which depends on $\mathbf{W}$ and $\mathbf{Z}$, and (iii) that the output $\mathbf{Y}$ of the activation function $\sigma$ is close enough to its input $\mathbf{WX}$, which depends $\sigma$ and $\mathbf{W}$, which is in turn determined by the learning process on the network. Notice that the condition $\delta > 0$ implicitly requires that $n > r$ (otherwise there is not even an $(r + 1)^{\text{th}}$ singular value). More than a requirement, this is a fundamental identifiability condition, because it takes $r_k$ linearly independent vectors to identify an $r_k$-dimensional subspace, and it takes $r = \sum_{k=1}^{K} r_k$ vectors to identify a union of K subspaces with dimensions $r_1, \ldots, r_K$. The spectral-gap assumption in 3.1 is introduced to establish sufficient conditions under which subspace clustering is guaranteed to be preserved. While such a gap may not hold uniformly across all real-world datasets, it provides a clean and interpretable theoretical mechanism linking perturbation stability to clustering structure.

There are four key players in Theorem 3.1: the data and the noise, which are beyond our control, the activation function $\sigma$, which we are free to choose, and the parameters $\mathbf{W}$, which are learned by the network. What makes this paper particularly interesting is that it is easy to construct matrices $\mathbf{W}$ that violate the conditions of Theorem 3.1 for any $\sigma$. Trivial examples include matrices whose product with $\mathbf{X}$ yields many negative values to be replaced with zeros in a ReLU transformation, destroying the subspace structure in the original data (see Figure 1). Naturally, one could use regularization techniques to encourage weights $\mathbf{W}$ that satisfy the conditions of Theorem 3.1. This could involve, for example, for ReLU activations, adding a penalty term that favors nonnegative weights $\mathbf{W}$ or nonnegative products $\mathbf{WX}$. The surprising part, as we will demonstrate below, is that neural networks tend to learn weights that inherently preserve the clustering structure even in the absence of explicit mechanisms enforcing this behavior.

# 4 PROOF

The proof of Theorem 3.1 is presented in three parts, showcased in Figure 1. First, we show that the clustering structure of $\mathbf{X}^\star$ can be estimated in closed-form. Then we establish that such estimator is invariant under arbitrary linear transformations. Finally, we demonstrate that for certain matrices $\mathbf{W}$, the non-linear transformation induced by certain activation functions do not disrupt the clustering structure of our closed-form estimator.

## 4.1 ESTIMATING THE GENERAL CLUSTERING STRUCTURE

The key observation is that the features of $\mathbf{X}^\star$ lie in a subspace whose projection operator encodes the clustering, which in turn will determine $\{\mathbf{U}_k\}_{k=1}^{K}$ and $\{\mathbf{v}_i\}_{i=1}^{n}$ (up to a basis rotation). To see this, define $n_k := |\mathbf{\Omega}_k|$ as the number of columns in $\mathbf{X}^\star$ corresponding to the $k^{\text{th}}$ subspace, and let $\mathbf{X}_k^\star$ be the $m \times n_k$ matrix containing such columns. For our analysis, assume without loss of generality that $\mathbf{X}^\star = [\mathbf{X}_1^\star, \ldots, \mathbf{X}_K^\star]$ (otherwise simply multiply by a permutation matrix on the right). Next let $\mathbf{U} := [\mathbf{U}_1, \ldots, \mathbf{U}_K]$ be the $m \times r$ matrix containing the concatenation of the bases $\{\mathbf{U}_k\}_{k=1}^{K}$. In addition, define $\mathbf{V}_k$ as the $r_k \times n_k$ matrix whose columns are the coefficients of $\mathbf{X}_k$ with respect to $\mathbf{U}_k$, i.e., $\{\mathbf{v}_i\}_{i \in \mathbf{\Omega}_k}$. Finally, let $\mathbf{V}$ be the $r \times n$ block-diagonal matrix whose diagonal blocks are $\{\mathbf{V}_k\}_{k=1}^{K}$. Then $\mathbf{V}$ has the following group-sparse structure, where the white areas represent zeros:

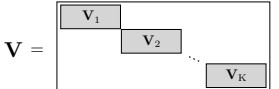

However, directly recovering $\mathbf{V}$ is challenging, since subspaces can admit multiple equivalent bases. To obtain a unique representation of the clustering structure, we define the projection operator onto the row space of the data

This way, we can rewrite (1) as $\mathbf{X} = \mathbf{U}\mathbf{V} + \mathbf{Z}$. Notice that $\mathbf{V}$ encodes the clustering structure, because if the $\mathrm{i}^{\mathrm{th}}$ column of $\mathbf{V}$ is nonzero on the $\mathrm{k}^{\mathrm{th}}$ block, then $\mathbf{x}_i^\star$ belongs to the $\mathrm{k}^{\mathrm{th}}$ cluster. Unfortunately, learning the specific basis $\mathbf{V}$ can be difficult, even if $\mathbf{Z} = \mathbf{0}$, because subspaces have infinitely many bases, most of which do not have the group-sparse structure of $\mathbf{V}$. However, we can still estimate the clustering structure in $\mathbf{V}$ through the projection operator onto its row space. To see this, let $\{\bar{\mathbf{V}}_k\}_{k=1}^K$ be orthonormal bases with the same spans as $\{\mathbf{V}_k\}_{k=1}^K$, and use $\{\bar{\mathbf{V}}_k\}_{k=1}^K$ instead of $\{\mathbf{V}_k\}_{k=1}^K$ to construct $\bar{\mathbf{V}}$. That is, $\bar{\mathbf{V}}$ is the $\mathrm{r} \times \mathrm{n}$ block-diagonal matrix whose diagonal blocks are $\{\bar{\mathbf{V}}_k\}_{k=1}^K$. This way, $\bar{\mathbf{V}}$ has the same group-sparse structure as $\mathbf{V}$. Since the row-blocks of $\bar{\mathbf{V}}$ are disjoint, it follows by construction that $\bar{\mathbf{V}}$ is orthonormal and spans the same subspace as $\mathbf{V}$. Recall that given a matrix, $\mathbf{P}(\cdot)$ denotes the projection operator onto its principal $\mathrm{r}$-dimensional row-space. Since projection operators are unique, as long as $\mathrm{r} < \mathrm{n}$,

$$\mathbf{P}(\mathbf{X}^\star) = \mathbf{P}(\bar{\mathbf{V}}) = \bar{\mathbf{V}}^\mathsf{T}\bar{\mathbf{V}}. \tag{3}$$

In words, the condition $\mathrm{r} < \mathrm{n}$ requires that the sum of the dimensions of the K subspaces is smaller than the total number of samples, which is an information-theoretic requirement for clustering (Pimentel-Alarcon & Nowak, 2016). From (3) we can see that the columns and rows of $\mathbf{P}(\mathbf{X}^\star)$ have the exact same support as the rows in $\bar{\mathbf{V}}$ (and $\mathbf{V}$), and have the following general structure:

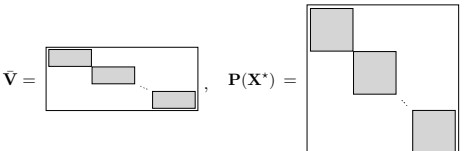

We thus see that learning $\mathbf{P}(\mathbf{X}^\star)$ is just as effective as learning $\mathbf{V}$ or $\bar{\mathbf{V}}$ in the sense that it encodes the clustering of $\mathbf{X}^\star$, because if the $(\mathrm{i}, \mathrm{j})^{\mathrm{th}}$ entry of $\mathbf{P}(\mathbf{X}^\star)$ is nonzero, then the $\mathrm{i}^{\mathrm{th}}$ and $\mathrm{j}^{\mathrm{th}}$ columns of $\mathbf{X}^\star$ correspond to the same cluster. Therefore, if we can recover the support of $\mathbf{P}(\mathbf{X}^\star)$, we obtain the clustering of $\mathbf{X}^\star$, as desired. Fortunately, since projection operators are unique, $\mathbf{P}(\mathbf{X}^\star)$ can be directly estimated as $\mathbf{P}(\mathbf{X})$. The following lemma states that under general conditions on the noise $\mathbf{Z}$, the difference between $\mathbf{P}(\mathbf{X}^\star)$ and $\mathbf{P}(\mathbf{X})$ is bounded.

**Lemma 4.1.** *Let $\delta_1 > 0$ be the gap between the $\mathrm{r}^{\mathrm{th}}$ singular value of $\mathbf{X}^\star$ and the $(\mathrm{r}+1)^{\mathrm{th}}$ singular value of $\mathbf{X}$. Suppose*

$$\delta_1 > \sqrt{2^7\mathrm{r}}\|\mathbf{Z}\|/\epsilon =: \eta_1, \tag{4}$$

*where $\epsilon > 0$ denotes the smallest absolute value in the support of $\mathbf{P}(\mathbf{X}^\star)$. Then*

$$\|\mathbf{P}(\mathbf{X}) - \mathbf{P}(\mathbf{X}^\star)\|_\infty < \epsilon/8. \tag{5}$$

Recall that $\delta_1$ can be interpreted as the similarity between the subspaces spanned by $\mathbf{X}^\star$ and $\mathbf{X}$. The condition in (4) essentially requires that the noise $\mathbf{Z}$ is not too large relative to the variance of $\mathbf{X}^\star$, so that the clusters are discernible from our estimator $\mathbf{P}(\mathbf{X})$, and no sample is misclustered.

*Proof.* We will show that corresponding entries in $\mathbf{P}(\mathbf{X}^\star)$ and $\mathbf{P}(\mathbf{X})$ cannot differ by more than $\epsilon/8$. To see this, write

$$
\begin{aligned}
\|\mathbf{P}(\mathbf{X}) - \mathbf{P}(\mathbf{X}^\star)\|_\infty^2 \;&\leq\; \|\mathbf{P}(\mathbf{X}) - \mathbf{P}(\mathbf{X}^\star)\|_F^2 \\
&= \; \|\mathbf{P}(\mathbf{X})\|_F^2 + \|\mathbf{P}(\mathbf{X}^\star)\|_F^2 - 2\mathrm{tr}(\mathbf{P}(\mathbf{X})^\mathsf{T}\mathbf{P}(\mathbf{X}^\star)) \\
&= \; 2\mathrm{r} - 2\mathrm{tr}(\mathbf{P}(\mathbf{X})^\mathsf{T}\mathbf{P}(\mathbf{X}^\star)) \;=\; 2(\mathrm{r} - \|\mathbf{P}(\mathbf{X})^\mathsf{T}\mathbf{P}(\mathbf{X}^\star)\|_F^2) \\
&=: \; 2(\mathrm{r} - \|\cos^2(\boldsymbol{\Theta})\|_F^2) \;=\; 2\|\sin^2(\boldsymbol{\Theta})\|_F^2 \;\leq\; 2\|\mathbf{Z}\|_F^2/\delta_1^2,
\end{aligned}
$$

where the last inequality follows directly by the Davis-Kahan $\sin(\boldsymbol{\Theta})$ Theorem (Davis & Kahan, 1970; Stewart & Sun, 1990). Then

$$
\|\mathbf{P}(\mathbf{X}) - \mathbf{P}(\mathbf{X}^\star)\|_\infty \;\leq\; \frac{\sqrt{2}\|\mathbf{Z}\|_F}{\delta_1} \;\leq\; \frac{\sqrt{2\mathrm{r}}\|\mathbf{Z}\|}{\delta_1}.
$$

Substituting $\delta_1$ from (4), we see that

$$
\|\mathbf{P}(\mathbf{X}) - \mathbf{P}(\mathbf{X}^\star)\|_\infty \;\leq\; \frac{\sqrt{2\mathrm{r}}\|\mathbf{Z}\|}{\delta_1} \;<\; \frac{\sqrt{2\mathrm{r}}\|\mathbf{Z}\|\epsilon}{\sqrt{2^7\mathrm{r}}\|\mathbf{Z}\|} \;=\; \frac{\epsilon}{8}.
$$

$\square$

## 4.2 Invariance to linear transformations

We will now show that linear transformations preserve clustering structure. More precisely, we will show that under reasonable conditions, the projection matrices of $\mathbf{X}$ and $\mathbf{W}\mathbf{X}$ are sufficiently close to one another and so they share the same clustering structure. This is summarized in the following lemma:

**Lemma 4.2.** *Suppose the gap $\delta_2 > 0$ between the $\mathrm{r}^{\mathrm{th}}$ singular value of $\mathbf{W}\mathbf{X}^\star$ and the $(\mathrm{r}+1)^{\mathrm{th}}$ singular value of $\mathbf{W}\mathbf{X}$ satisfies $\delta_2 > \sqrt{2^7\mathrm{r}}\|\mathbf{W}\mathbf{Z}\|/\epsilon$. Then $\|\mathbf{P}(\mathbf{X}) - \mathbf{P}(\mathbf{W}\mathbf{X})\|_\infty < \epsilon/4$.*

*Proof.* Start with two triangle inequalities:

$$
\begin{aligned}
\|\mathbf{P}(\mathbf{X}) - \mathbf{P}(\mathbf{W}\mathbf{X})\| \;&\leq\; \|\mathbf{P}(\mathbf{X}) - \mathbf{P}(\mathbf{X}^\star)\| + \|\mathbf{P}(\mathbf{X}^\star) - \mathbf{P}(\mathbf{W}\mathbf{X}^\star)\| + \|\mathbf{P}(\mathbf{W}\mathbf{X}^\star) - \mathbf{P}(\mathbf{W}\mathbf{X})\| \\
&= \; \|\mathbf{P}(\mathbf{X}) - \mathbf{P}(\mathbf{X}^\star)\| + \|\mathbf{P}(\mathbf{W}\mathbf{X}^\star) - \mathbf{P}(\mathbf{W}\mathbf{X})\|,
\end{aligned} \tag{6}
$$

where the last equality follows because as long as $\mathrm{rank}(\mathbf{W}) \geq \mathrm{r}$ (implicitly required by the theorem), then $\mathbf{P}(\mathbf{X}^\star) = \mathbf{P}(\mathbf{W}\mathbf{X}^\star) = \mathbf{V}^\mathsf{T}\mathbf{V}$. Using the exact same arguments as in the proof of Lemma 4.1, we can bound

$$
\|\mathbf{P}(\mathbf{W}\mathbf{X}^\star) - \mathbf{P}(\mathbf{W}\mathbf{X})\|_\infty^2 \;\leq\; \frac{\sqrt{2\mathrm{r}}\|\mathbf{W}\mathbf{Z}\|}{\delta_2} \;<\; \frac{\sqrt{2\mathrm{r}}\|\mathbf{W}\mathbf{Z}\|\epsilon}{\sqrt{2^7\mathrm{r}}\|\mathbf{W}\mathbf{Z}\|} \;=\; \frac{\epsilon}{8}.
$$

Plugging this and (5) in (6) we obtain the lemma. $\square$

## 4.3 Invariance to certain activation functions

Finally, we specify that under certain conditions on $\sigma$ and $\mathbf{W}$, the clustering structure of $\mathbf{W}\mathbf{X}$ is the same as that of $\mathbf{Y}$.

**Lemma 4.3.** *Let $\delta_3 > 0$ be the gap between the $\mathrm{r}^{\mathrm{th}}$ singular value of $\mathbf{Y}$ and the $(\mathrm{r}+1)^{\mathrm{th}}$ singular value of $\mathbf{W}\mathbf{X}$. Suppose $\delta_3 > \sqrt{2^7\mathrm{r}}\|\mathbf{Y} - \mathbf{W}\mathbf{X}\|/\epsilon$. Then $\|\mathbf{P}(\mathbf{Y}) - \mathbf{P}(\mathbf{W}\mathbf{X})\|_\infty \leq \epsilon/8$.*

The proof of Lemma 4.3 follows by the same arguments in Lemma 4.1. The proof of Theorem 3.1 follows directly by Lemmas 4.1, 4.2, and 4.3, and three triangle inequalities.

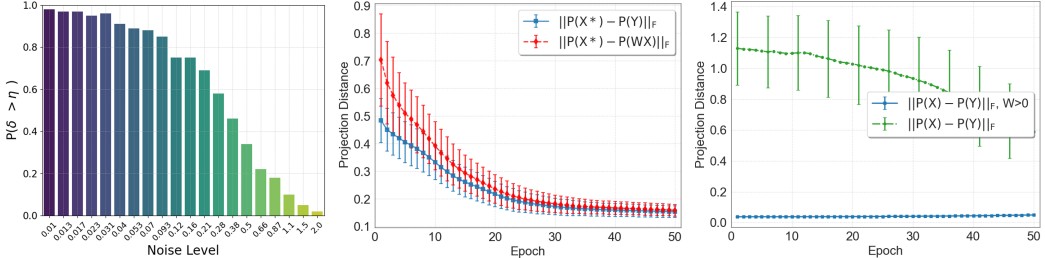

Figure 2: **Left:** Probability with which the conditions in Theorem 3.1 hold, guaranteeing that subspace structure is preserved through a ReLU transformation. **Center:** Projection distances as a function of epochs with noise $s^2 = 0.1$, showing that the network is inherently learning weights that preserve clustering structure. **Right:** Projection distances for two different initializations. The initialization informed by our analysis ($\mathbf{W} > \mathbf{0}$) perfectly preserves clustering structure and is a local optima of the learning process (its gradient is numerically zero and its hessian determinant is positive).

## 5 INHERENT PRESERVATION OF SUBSPACE CLUSTERING STRUCTURE

According to Theorem 3.1, several conditions must align so that the cluster structure in $\mathbf{X}^\star$ is preserved in $\mathbf{Y}$. First, the subspace signal in $\mathbf{X}^\star$ must overcome the noise $\mathbf{Z}$, as required by the bound on $\delta(\mathbf{X}^\star, \mathbf{X})$. Next, the transformation $\mathbf{W}$ must not blow up the noise $\mathbf{Z}$, as required by the bound on $\delta(\mathbf{W}\mathbf{X}^\star, \mathbf{W}\mathbf{X})$. Finally, the activation function must not disrupt the cluster structure in $\mathbf{W}\mathbf{X}$, as required by the bound on $\delta(\mathbf{Y}, \mathbf{W}\mathbf{X})$. The first condition (on the data and the noise) is entirely out of our control, and must be assumed. On the other hand, given the data, the second and third conditions depend entirely on the choice of $\sigma$ and $\mathbf{W}$, which as Figure 1 shows, may preserve the subspace structure in $\mathbf{X}^\star$, or destroy it entirely.

Characterizing when the conditions in Theorem 3.1 will hold can be difficult in practice, as $\mathbf{X}^\star$ is generally unknown, and $\mathbf{W}$ is the parameter to be learned by the training process (and our assumptions depend on both these quantities). As discussed earlier, we could regularize our objective function to encourage weights $\mathbf{W}$ that meet the conditions of Theorem 3.1. This could be achieved by penalizing negative entries in $\mathbf{W}$ or $\mathbf{W}\mathbf{X}$. The surprising part is that this seems to be unnecessary, as certain neural networks appear to favor such solutions inherently.

To see this we present a numerical experiment to quantify the frequency with which our conditions hold on synthetic data following the subspace clustering model. Specifically, we generated K $r_k$-dimensional subspaces of $\mathbb{R}^m$, each spanned by a matrix $\mathbf{U}_k \in \mathbb{R}^{r_k \times n}$ with i.i.d. entries drawn from the standard gaussian distribution $\mathcal{N}(0, 1)$, which we subsequently orthogonalized. We similarly populated $\mathbf{V}_k \in \mathbb{R}^{r_k \times n}$ with i.i.d. $\mathcal{N}(0, 1/m)$ entries and constructed $\mathbf{X}_k^\star = \mathbf{V}_k \mathbf{U}_k + \mathbf{Z}_k$ and $\mathbf{X}^\star = [\mathbf{X}_1^\star, \dots, \mathbf{X}_K^\star]$. Then we populated $\mathbf{Z} \in \mathbb{R}^{m \times n}$ with i.i.d. $\mathcal{N}(0, s^2)$ entries. Here $s^2$ represents the noise variance. Finally, we initialized $\mathbf{W}$ with i.i.d. uniform entries in the range $(-m^{1/2}, m^{1/2})$, and proceeded to learn the parameters $\mathbf{W}$ of a single hidden feedforward layer autoencoder with 80 ReLU neurons using standard gradient descent with a learning rate of $0.005$. We trained this autoencoder using squared Frobenius reconstruction loss $\|\mathbf{X} - \hat{\mathbf{X}}\|_F^2$. For the dataset, we used $r_k = 4$, $m = 400$, $n_k = 100$, and $K = 4$, so that $n = 400$. Then we proceeded to calculate $\delta$ and $\eta$ as a function of the noise variance $s^2$, which is a proxy of $\|\mathbf{Z}\|$ in (2), and recorded the frequency with which our assumptions hold (i.e., when $\delta > \eta$). All experiments were conducted on a computer with an AMD Ryzen 7 5800H CPU, 16 GB RAM, and an NVIDIA GTX 1660 Ti GPU (6 GB). Figure 2-Left summarizes the results of 100 independent trials for each value of $s^2$. The results show that our bound degrades nicely with noise.

As Figure 1 shows, random weights will generally destroy the subspace clustering structure in $\mathbf{Y}$. Figure 2-Center shows that the network is inherently learning weights $\mathbf{W}$ that satisfy the conditions of our theorem, thus preserving the cluster structure in $\mathbf{Y}$. The surprising part is that it is doing so without any mechanism explicitly enforcing this behavior.

**Practical insights.** Beyond theory, our findings have practical gains. Notice that the conditions of Theorem 3.1 are generally met for ReLUs whenever $\mathbf{W}\mathbf{X} > \mathbf{0}$, because then $\sigma(\mathbf{W}\mathbf{X}) = \mathbf{W}\mathbf{X}$. This

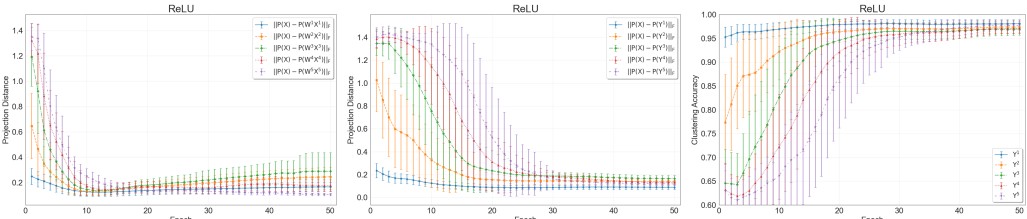

Figure 3: Evolution of clustering structure over epochs in a 5-layer feedforward ReLU network. **Left:** Projection distance between the input $\mathbf{X}^\ell$ and linear transformation $\mathbf{W}^\ell \mathbf{X}^\ell$ at each layer. **Center:** Projection distance between the input $\mathbf{X}^\ell$ and the ReLU output $\mathbf{Y}^\ell$ at each layer. These distances can be seen as a loss of clustering structure. Their decrease over epochs indicate that the network is learning weights that preserve such structure. **Right:** Clustering accuracy at each layer over epochs, obtained from the projection operator as described above. Summary of 100 independent trials.

is trivially true whenever $\mathbf{X} > \mathbf{0}$ and $\mathbf{W} > \mathbf{0}$. In many cases, $\mathbf{X}$ is nonnegative due to the nature of many modern datasets. Examples arise in network inference (Eriksson et al., 2011), single-cell sequencing (Ding et al., 2006), drug discovery (Zhang et al., 2019), multi-omics (Chevrette & Currie, 2019), medical image processing (Riaz et al., 2020), and more (Huo et al., 2021). For example, hop counts in networked systems are nonnegative, pixel intensities are nonnegative, many biomedical features like age, heart rate, blood pressure, body temperature, glucose, cholesterol, oxygen saturation, and enzime levels, white blood cell count, respiratory rate, etc., are nonnegative. On the other hand, $\mathbf{W}$ may be initialized with nonnegative values ensuring that the clustering structure in the data is automatically preserved at the beginning of the training process. This simple strategy leads to practical gains in terms of accuracy and length of the learning process. To see this we repeated the same experiments as in Figure 2-Center, except that we increased the noise to $0.5$, and initialized $\mathbf{W}$ with i.i.d. entries in the unit interval. The results are in Figure 2-Right, where we can see that this initialization perfectly preserves clustering structure and is a local optima of the learning process that improves over other initializations.

## 6 Beyond a single hidden layer

Our analysis for a single layer can be directly extended to multiple layers by applying a union bound on Theorem 3.1. Let $\mathbf{X}^\ell$, $\mathbf{W}^\ell$, and $\mathbf{Y}^\ell$ denote the input, weights, and output of a network at the $\ell^{\text{th}}$ layer. The subspace clustering structure will be preserved at the $\ell^{\text{th}}$ layer if $\mathbf{P}(\mathbf{Y}^\ell)$ is close enough to $\mathbf{P}(\mathbf{X}^\star)$. This will be the case if the network learns parameters $\mathbf{W}^1, \dots, \mathbf{W}^\ell$ that simultaneously satisfy the conditions of the union-bounded version of Theorem 3.1 (with $\epsilon$ factored by $\ell$). Most surprisingly, deep networks display the same behavior observed in their shallow counterparts, and inherently preserve the subspace clustering structure in the data through multiple layers, in the absence of explicit mechanisms enforcing this behavior.

To demonstrate this, we replicate the exact same experiment as in Figure 2-Center, except that this time we used a network with 5 ReLU layers. Figure 3 summarizes the results of the training process, showing that the network starts with random weights that violate the assumptions of Theorem 3.1, destroying the clustering structure. As part of its training process it learns parameters that preserve the clustering structure encoded in the projection matrices.

## 7 Beyond ReLU activations

So far we have supported our conclusions with ReLU activations. However, there is nothing specific in our analysis that binds us to such activations. To demonstrate this we present a series of experiments with other activation functions and other architectures. Figure 4 shows the results for the same architecture as in Figure 3, except with for GELU (Hendrycks & Gimpel, 2023) and SiLU (Elfwing et al., 2017) activation functions, as well as a 3-layer LSTM architecture (Staudemeyer & Morris, 2019) with ReLU activations. In these experiments we generated data with the same procedure

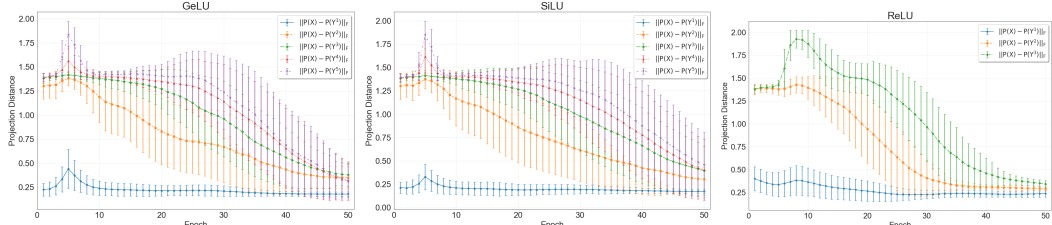

Figure 4: Evolution of clustering structure over epochs in 5-layers feedforward networks using GELU (left) and SiLU (center) activation functions, and a 3-layer LSTM architecture with ReLU activations (right). The decrease in the projection distances over epochs indicates that the networks are learning weights that preserve the clustering structure. This shows that clustering preservation is not an exclusive property of ReLUs or feed forward layers. Summary of 100 independent trials.

described in Figure 2-Center. This shows that clustering preservation is not an exclusive property of ReLUs or feedforward layers.

There are, of course, limitations to Theorem 3.1. It is easy to see that Theorem 3.1 cannot be easily extended to more sophisticated transformations that do not rely on a linear transformation and an activation function. These complex transformations are more likely to disrupt the clustering structure of the data. Examples include the convolution operator (O'Shea & Nash, 2015) or the Transformer (Vaswani et al., 2023). To demonstrate this we repeat the same experiments as in Figure 4, except using a 3-layer CNN and a 4-layer Transformer with different activation functions. The results are in Figure 5, showing that unlike linear operators, the convolution or attention transformations do not inherently preserve clustering structure.

**Remark.** We point out that Transformers and CNNs are *not* failing at clustering. They are still clustering accurately, same as all other networks. In fact, all models were intentionally selected because of their high clustering accuracy. We wanted models that cluster correctly because the goal of these experiments (and the paper) is not to analyze accuracy or establish a new state-of-the-art. Rather, we seek to investigate the behavior of deep networks, and understand the mechanisms they are using for clustering. We are focusing on projection distances because, beyond accuracy, they reveal that ReLU-type networks are clustering by learning the closed-form solution. The large projection distances exhibited by Transformers and CNNs show that these networks are clustering through some mechanism other than the closed-form solution, and such mechanism does not preserve the original clustering structure.

## 8    REAL DATA

We validate our conclusions across 12 diverse real-world datasets, including MNIST (Deng, 2012), CIFAR-10 (Krizhevsky, 2009), Extended MNIST (Cohen et al., 2017), Street View House Numbers (SVHN) (Netzer et al., 2011), Kuzushiji-MNIST (Clanuwat et al., 2018), Fashion MNIST (Xiao et al., 2017), Flowers-102 (Nilsback & Zisserman, 2008), Food-101 (Bossard et al., 2014), USPS Handwritten Digits (Hull, 1994), STL-10 (Coates et al., 2011), Oxford DTD dataset (Cimpoi et al., 2014), Oxford Pet dataset, and EuroSAT dataset (Helber et al., 2019) for the 5-layer network. The results and more details are presented in Figures 6 - 9 in the Appendix. We emphasize that the projection distances reported throughout the paper are not intended as clustering performance metrics. Rather, they measure the preservation of subspace structure during the learning process.

In all experiments considered, the models achieve perfect or near-perfect clustering accuracy under standard metrics such as ACC, NMI, and ARI. These models were intentionally selected because our goal is not to compare clustering performance, but to analyze how successful models achieve clustering. Our findings suggest that ReLU-type networks cluster by implicitly learning the closed-form projection structure described in Section 4.

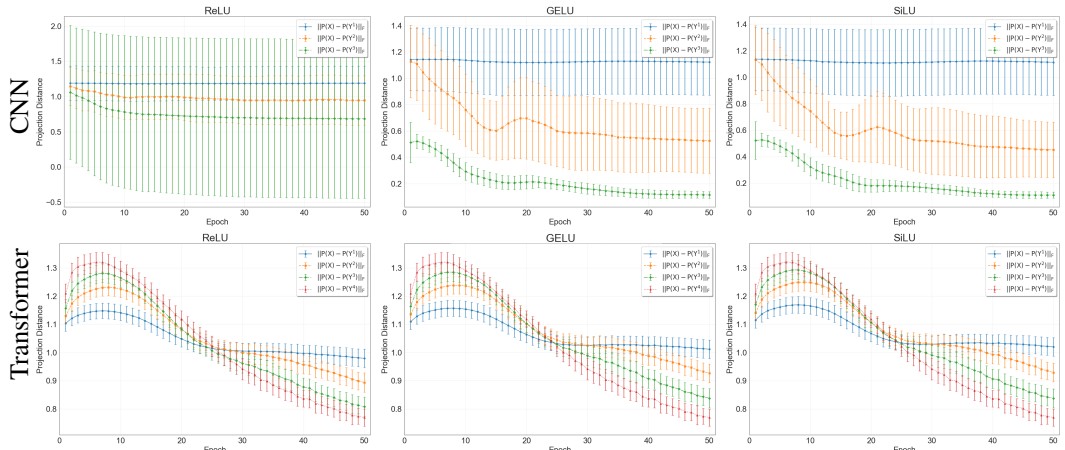

Figure 5: Evolution of projection distances over epochs with different activation functions. **Top:** 3-layers CNNs. **Bottom:** Transformers. These distances can be interpreted as a loss of clustering structure, showing that unlike linear operators, the convolution or Transformer do not inherently preserve clustering structure. Summary of 100 independent trials.

ACKNOWLEDGMENTS

This work was partially supported by NSF's CAREER Award #2239479.

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

# A APPENDIX A

The following section depicts experiments using 5 layer neural network for real data. The datasets included MNIST Deng (2012), CIFAR-10 Krizhevsky (2009), Extended MNIST Cohen et al. (2017), Street View House Numbers (SVHN) Netzer et al. (2011), Kuzushiji-MNIST Clanuwat et al. (2018), Fashion MNIST Xiao et al. (2017), Flowers-102 Nilsback & Zisserman (2008), Food-101 Bossard et al. (2014), USPS Handwritten Digits Hull (1994), STL-10 Coates et al. (2011), Oxford DTD dataset Cimpoi et al. (2014), Oxford Pet dataset, and EuroSAT dataset Helber et al. (2019). Figure 6 includes the projection distance vs epochs plots for MNIST, CIFAR-10, and Extended-MNIST. For the experiment, the model was run for 200 epochs with a fixed learning rate for all the datasets included in the experiments.

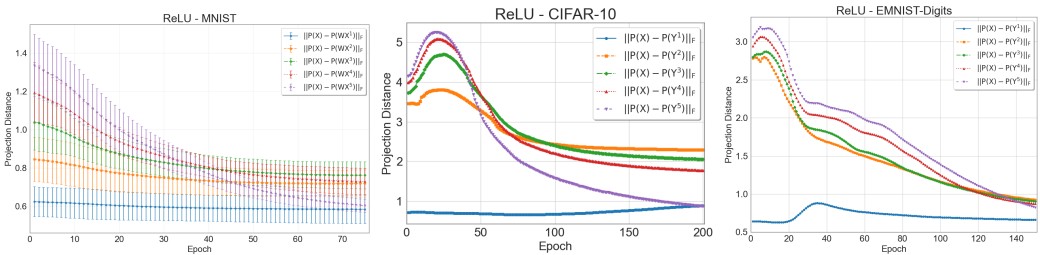

Figure 6: **Left:** MNIST dataset for 100 trials. **Center:** CIFAR-10. **Right:** Extended-MNIST

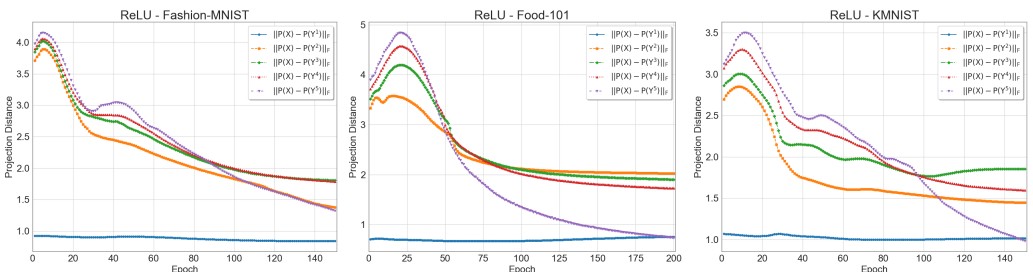

Figure 7: **Left:** Fashion-MNIST. **Center:** Food-101. **Right:** Kuzushiji-MNIST

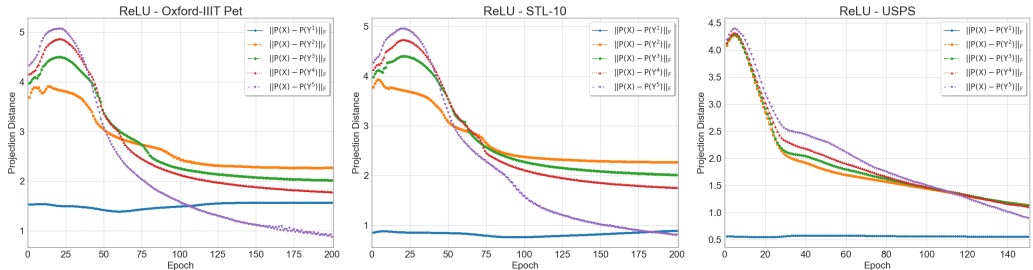

Figure 8: **Left:** Oxford Pet Dataset. **Center:** STL10 Dataset. **Right:** USPS Dataset

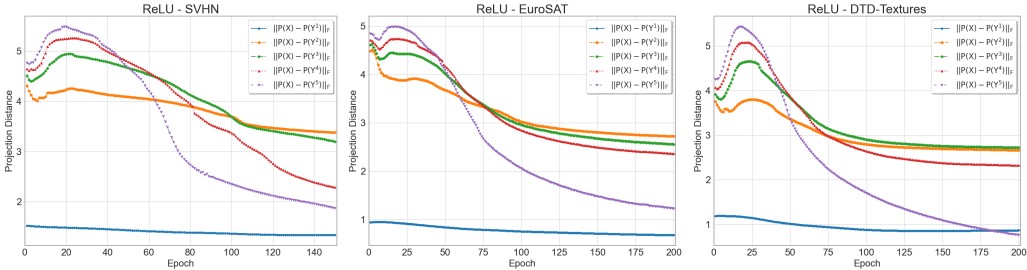

Figure 9: **Left:** SVHN. **Center:** EuroSAT Dataset. **Right:** Oxford DTD Textures Dataset

# B    APPENDIX B

**Multilayer Extension.**   The multi-layer bound mentioned in section 6 can be obtained with a triangle inequality. Specifically,

$$
\begin{aligned}
\|\mathbf{P}(\mathbf{X}) - \mathbf{P}(\mathbf{Y}^L)\| &= \|\mathbf{P}(\mathbf{X}) - \mathbf{P}(\mathbf{Y}^1) + \mathbf{P}(\mathbf{Y}^1) - \mathbf{P}(\mathbf{Y}^2) + \mathbf{P}(\mathbf{Y}^2) - \cdots + \mathbf{P}(\mathbf{Y}^{L-1}) - \mathbf{P}(\mathbf{Y}^L) \\
&\leq \|\mathbf{P}(\mathbf{X}) - \mathbf{P}(\mathbf{Y}^1)\| + \|\mathbf{P}(\mathbf{Y}^1) - \mathbf{P}(\mathbf{Y}^2)\| + \|\mathbf{P}(\mathbf{Y}^2) - \mathbf{P}(\mathbf{Y}^3)\| + \cdots + \|\mathbf{P}(\mathbf{Y}^{L-1}) - \mathbf{P}(\mathbf{Y}^L)\| \\
&< L\frac{\varepsilon}{2}.
\end{aligned}
$$

And the spectral gap must be the maximum among the corresponding factors of all layers analogous to those in equation 2.

