# OpenReview forum: "Some Neural Networks Inherently Preserve Subspace Clustering Structure"
_ICLR.cc/2026/Conference — ICLR 2026 Poster_

### Official Review · Reviewer_fY3w · 2025-10-29

**Soundness:** 3
**Presentation:** 3
**Contribution:** 3
**Rating:** 6
**Confidence:** 3

**Summary:**

This paper theoretically investigates the ability of certain neural networks to preserve subspace clustering structures and formalizes this conjecture using empirical research.

**Strengths:**

1. The paper offers a novel perspective, connecting the empirical success of neural networks to the classical theory of subspace clustering.
2. The writing is generally clear. The proofs are well-structured and convey the core ideas.
3. The experimental design is comprehensive, progressing from synthetic data to diverse real-world datasets, supporting the arguments.
4. It provides a viable explanation for the effectiveness of deep learning on data with inherent clustering structure.

**Weaknesses:**

1. The theoretical extension from single-layer to multi-layer networks is overly simplified. Section 6 claims to extend the results to multi-layer networks, but this is not fully elaborated.
2. The paper states that CNNs/Transformers achieve high accuracy without maintaining the original structure. It is worth exploring in depth the circumstances under which maintaining the original structure is preferable to reconstructing a new one.
3. The paper does not clearly specify the specific loss function and architectural details used for training the neural network.

**Questions:**

1. What specific training objectives did you use in the deep network, LSTM, and real-data experiments in Sections 6, 7, and 8? Was it autoencoders and mean squared error loss, or something else?
2. For the multi-layer extension in Section 6, can it provide a rigorous mathematical formulation or derivation of how Theorem 3.1 applies to $L$ layers via the joint bound? What is the resulting total error bound, and what requirements does it impose on the error $\epsilon_\ell$ for each layer?
3. The theory assumes additive noise Z. Has the conclusion been explored for more complex, non-additive, or data-dependent noise?

---

> ### Author Response · Authors · 2025-11-14
> **Author's response to reviewer fY3w**
>
> We thank the reviewer for the thoughtful and encouraging evaluation. We also thank the reviewer for raising insightful questions, which we address below:
>
> 1. The details about number of neurons, layers, and learning parameters can be found in the paragraph starting in line 345. We can include a Table in the Appendix with the details of each model and dataset to make it easier to consult. In the same paragraph we mention that we used a standard auto encoder, but we can explicitly state our loss function, which you are correct, is the squared loss || X - Xhat ||_F.
>
> 2. The multi-layer bound can be obtained with a triangle inequality. Specifically, ||P(X)-P(Y^L)| = ||P(X)-P(Y^1) + P(Y^1) - P(Y^2) + P(Y^2) + … + P(Y^L)-P(Y^L) || <= ||P(X)-P(Y^1)|| + ||P(Y^1) - P(Y^2)|| + ||P(Y^2)-P(Y^3) || + … + || P(Y^L)-P(Y^L) || < L epsilon/2, and the spectral gap must be the maximum among the corresponding factors of all layers analogous to those in equation (2). We can certainly include this in an appendix in the final version.
>
> 3. Our theory does not exclude data-dependent noise, as long as the spectral gap condition is preserved. Naturally, data-dependent noise or adversarial noise can be easily constructed to break such condition. That said, you are correct that our current derivations only apply to additive noise (which relies on additive perturbation theory). Generalizing it to more complex, non-additive noise would require an entirely different analysis that may require other perturbation theory tools, or entirely different tools.  These other types of non-additive noises are therefore out of the scope of this paper, but are definitely interesting lines for future work.
>
> We hope that our clarifications address the reviewer's concerns, and hope that they consider increasing their score. We will be happy to clarify further if necessary.

---

### Official Review · Reviewer_R2m9 · 2025-10-29

**Soundness:** 2
**Presentation:** 2
**Contribution:** 2
**Rating:** 2
**Confidence:** 4

**Summary:**

The paper attempts to develop theoretical understanding on the observation of some neural networks preserving clustering structure, in which precise conditions for clustering structure preserving is provided. Some theoretical results are developed to bound the gap beween the projected differences between the clean data, transformed data and the output data after one-layer transform. Moreover, numerical analysis and empirical evaluation on real world data are shown, with some explaination and discussions.

**Strengths:**

+ It is appealing to investigate the observation of clustering structure preserving with some theoretical justification.

**Weaknesses:**

1. The mathematical analysis in the paper is confusing. Symbols used in proofs are not clearly defined or without any interpretation. For example, the norm of the matrix, $\| Z\|$ and $\| Z \|_\infty$ are not clearly defined, $\sin (\Theta)$ is not clearly defined. Even worse, the projection $P$ and $W$ are also not clear.

2. The derivations in the proofs are incorrect.
- For example, in the proof of Lemma 4.1, the third equality seems incorrect. How to build the equality from a trace of the two projected matrix to the squared Frobenius norm of them?
- In the proof of Lemma 4.2, the reviewer cannot see how "plugging this and (5) in (6) we obtain the lemma." The reviewer cannot obtain the result in Lemma 4.2. The proof for Lemma 4.2 are wrong.  Moreover, for the matrix $W$, it is weight matrix, without any implicit assumption on the rank (if not specially claimed).

3. Regarding to Theorem: It is not clear how the condition of $\delta_1$ which is defined as the singular value gap of $X$ and $X^\ast$ is actually used.

4. The empirical evaluations are insufficient and incomplete. How the neural networks are trained actually? Which loss function is used? Are the empirical results obtained from the synthetic data generalizable in a broad sense, or it is just because of the special structure of the synthetic data? The way to generate the synthetic data is to form a set of nearly orthogonal subspaces.

5. Is the initialization of $W$ reasonable? What happens if $W$ is initialized with an iid Gaussian of zero mean and variance of $1/m$? From the viewpoint of the reviewer, it is more natural and making sense, or even with nice property.  Why initialize $W$ with i.i.d. uniform entries in the range of $(- \sqrt(m), \sqrt(m))$?

6. The worse results in Fig.2 (right panel) are not clearly interpreted.  In Fig. 5, why CNN and Transformers are failed to preserve the clustering structure? How the clustering accuracy is obtained in Fig.4?

7. Minor issues:
- The format for the citation in some places seems not properly used.
- The first three paragraphs of the introduction read like generated automatically via a LLM. Some contents are not factual.

**Questions:**

Please read the weaknesses.

---

> ### Author Response · Authors · 2025-11-12
> **Author rebuttal to Reviewer R2m9**
>
> We thank the reviewer for their comments. We are happy to clarify each item:
>
> 1) The norms we employ (spectral, Frobenius, and infinity), along with the corresponding notations we use for them, are standard and widely recognized within ICLR’s target audience as well as in standard linear algebra references. For this reason, we did not consider it necessary to define them explicitly. Nevertheless, we would be happy to include an appendix providing these definitions for completeness. The definition of sin(Theta), obtained through a basic trigonometric identity of cos(Theta), defined through the principal angles between two subspaces, is in line 265. We are also happy to include these definitions in the appendix. The precise definitions of P and W are in lines 135 and 138.
>
> 2) We believe that our proofs are correct: the 3rd equality in Lemma 4.1 follows by using some basic trace manipulations and recalling that P(X) and P(X*) are projection matrices, which are idempotent and symmetric:
> $
> || P(X)' P(X*) ||^2_F := tr[ (P(X)' P(X*))' P(X)' P(X*) ] = tr[ P(X*)' P(X) P(X)' P(X*) ] = tr[ P(X) P(X)' P(X*) P(X*)' ] = tr[ P(X)'P(X)' P(X*) P(X*) ]
> = tr[ P(X)' P(X*) ]
> $
> We did not consider this step-by-step granularity necessary, but we are happy to include it in the final version. As for Lemma 4.2, it is simply a substitution:
>
> $
> || P(X)-P(X*)|| < \\epsilon/8
> $
> ,
> $
> || P(WX*)-P(WX)|| < \\epsilon/8
> $
> , therefore the sum of these two factors in equation (6) is $<\\epsilon/4$, which is the result of Lemma 4.2.
>
> 3) The condition on $\\delta$ is the assumption of the Theorem. As to how to use it, well, we simply check: if the assumption is met, then the Theorem holds, otherwise, the Theorem doesn't hold.
>
> 4) All details about the synthetic (standard gaussian) data generation, network parameters, learning rates, etc. are in the paragraph starting in line 345. These simulations are complemented with 12 diverse real-world datasets. Based on accepted papers at ICLR, ICML, NeurIPS, etc., we believe this level of experimentation is significantly above average for this type of paper, which is not an exhaustive comparative survey paper.
>
> 5) There are many acceptable ways to initialize W. We can include the one that you suggest as well as several other reasonable ones in an appendix.
>
> 6) We are not interpreting any result in Figure 2-right. We are simply stating facts: (i) our initialization perfectly preserves clustering structure (which is a measurable and verifiable fact evidenced by the figure), and (ii) our initialization is a local optima of
> the learning process -- which again is a fact, because its gradient is numerically zero and its hessian determinant is positive, as we clearly state. As to Figure 5, that is precisely the point of that figure: to raise the question "why do CNNs and Transformers fail to preserve clustering structure?", and open it up for future research, out of the scope of this paper, which is focused on ReLU-type networks (which do preserve clustering structure).
>
> 7) We will review all our referencing formatting. We are flattered that you think our writing has LLM quality. All our sentences were written by ourselves, with minor LLM assistance as a Thesaurus. All our statements are backed up by references. If you could be more specific as to what is incorrect or inaccurate, we will be happy to rephrase to improve clarity.
>
> We hope that our clarifications address the reviewer's concerns, and hope that they consider increasing their score. We will be happy to clarify further if necessary.

---

### Official Review · Reviewer_QkDR · 2025-11-01

**Soundness:** 2
**Presentation:** 3
**Contribution:** 3
**Rating:** 6
**Confidence:** 3

**Summary:**

The paper presents a theoretical analysis of when neural networks preserve clustering structure under a union of subspaces (UoS) model. It derives conditions ensuring that a network’s learned representations retain the subspace structure of clean data, even when inputs are noisy. Through a perturbation analysis, the authors show that the row space of the learned representation closely matches that of the original data. Experiments support the theory, revealing that certain networks naturally maintain clustering structure, offering insights into initialization, feature encoding, and regularization.

**Strengths:**

- The paper fills a theoretical gap by analysing the clustering behaviour of neural networks using subspace clustering and perturbation theory, offering a proof for a long-standing empirical intuition.
- Synthetic experiments demonstrate that gradient descent naturally enforces the theoretical condition without explicit regularisation.
- Provides insights for neural network design decisions, like initialisation

**Weaknesses:**

- The UoS model and large spectral-gap condition may not hold in realistic settings
- The paper relies only on projection-distance measures; evaluating standard clustering metrics (e.g., NMI, ARI) could show how clustering performance is linked to the network architecture and behaviour of projection-distances.
- The paper only briefly discusses that breaking subspace clustering structure, as seen in CNNs and Transformers, may actually contribute to their strong empirical performance. I found this insight actually very interesting, as these architectures are more commonly used in practice than MLPs.

**Questions:**

- How is the final clustering performance measured w.r.t. to the ground truth (e.g., with ACC, NMI or ARI) of a network linked to its preservation of subspace clustering structure? I imagine that clustering performance of CNN's or Transformers would generally be higher than the performance of MLPs
- In the paper you mention: "The large projection distances exhibited by Transformers and CNNs show that these networks are clustering through some mechanism other than the closed-form solution, and such mechanism does not preserve the original clustering structure." Could you elaborate on this? This is a very interesting insight, and I would be curious to see a more in depth discussion in the paper on this point.

---

> ### Author Response · Authors · 2025-11-14
> **Author response to Reviewer QkDR**
>
> We thank the reviewer for the thoughtful and encouraging evaluation. We also thank the reviewer for raising insightful questions, which we address below:
>
> 1. The spectral-gap condition is introduced to theoretically demonstrate that there exist conditions to guarantee subspace clustering preservation. While this condition may not be perfectly satisfied for all real datasets, our experiments show that some datasets indeed satisfy this condition (Figure 2-left). Moreover, our experiments show that the subspace-preserving structure remains stable under weaker conditions than our bound requires (naturally, as it is theoretical bound).
>
> 2. We clarify that the projection distances are not used as a clustering metric, but rather as a metric of structure preservation throughout the learning process. In fact, all models exhibit perfect or near perfect clustering accuracy in these rather simple tasks (under ACC, NMI or ARI). That is actually why they were specifically selected — because we are not focused in understanding which model clusters better. Rather, we want to understand how successful models cluster, and what is the mechanism that they use. We found that ReLU-type networks seem to use the closed-form structure-preservation mechanism. We will be glad to include a remark clarifying this in the final version. We can also include an appendix with clustering accuracy metrics (ACC, NMI, and ARI).
>
> 3. The observation that CNNs and Transformers do not preserve subspace clustering structure is only meant to (i) point out that not all architectures preserve this structure, and (ii) to open the question: since they are ‘preferring’ to ‘break’ such structure, what mechanism are they favoring instead? As you point out, this is indeed a very interesting question! We plan to explore it in our future work, as it is out of the scope of the current paper, which focuses specifically on ReLU-type networks. We will be happy to clarify this in the final version.
>
> We hope that our clarifications address the reviewer's concerns, and hope that they consider increasing their score. We will be happy to clarify further if necessary.

---

> > ### Comment · Reviewer_QkDR · 2025-11-19
> >
> > Thank you for addressing my questions. After reading your response and the response to other reviewers I decided to keep my initial score.

---

### Meta-Review · Area_Chair_EqNF · 2026-01-06

**Summary:**

This paper addresses a fundamental question in neural network theory: why certain architectures naturally preserve the clustering structure of data without explicit regularization. The authors formalize this through the lens of subspace clustering and provide theoretical bounds on structure preservation.

Two reviewers rated 6, citing the paper's novelty in connecting deep learning success to classical subspace clustering theory. They found the experimental design comprehensive, covering synthetic and real-world datasets. Reviewer R2m9 rated 2, citing confusing notation and incorrect proofs. However, the authors' rebuttal successfully demonstrated that the "incorrect" proofs were actually standard trace manipulations.

One concern not raised by any reviewer is a lack of connection to the community of work on understanding deep architectures under the union of subspace assumption, e.g. [White-Box Transformers via Sparse Rate Reduction](https://arxiv.org/abs/2306.01129), [Deep Self-expressive Learning](https://proceedings.mlr.press/v234/zhao24a/zhao24a.pdf)

Overall, the paper's contribution to understanding the "inherent" bias of ReLU-type networks toward preserving data geometry is valuable to the ICLR community. The authors have committed to including the expanded proofs and a table of architectural details in the final version to further improve clarity. I recommend an acceptance.

**Reviewer Concerns:**

Reviewer QKDR:

- The reviewer questioned if the UoS model and large spectral-gap conditions hold in real-world settings: Addressed. The authors clarified that while the theoretical bound is strict, experiments show the structure is stable under weaker conditions, and some real datasets do satisfy the spectral-gap requirement.

- Absence of metrics like NMI or ARI to link structure preservation to actual clustering performance: Addressed. Authors explained that because the chosen models all achieve near-perfect accuracy on these tasks, they focused on projection distance rather than performance. They offered to include an appendix with standard metrics.

- Why more complex architectures such as CNNs/Transformers "break" the subspace structure: Acknowledged as future work.

Reviewer R2m9:

This reviewer has several concerns on notations, technical errors, and experimental details. All were addressed in rebuttal.

Reviewer fY3w:

- Extension from single-layer to multi-layer networks overly simplified: Addressed. The authors provided a mathematical formulation using triangle inequality and the spectral gap of individual layers to derive a total error bound.

- Missing details on training loss and architecture: Addressed. Authors offered to add details to appendix.

- The reviewer asked if the theory applies to non-additive or data-dependent noise: Partially addressed. Data-dependent noise is technically covered if the spectral gap holds, but non-additive noise would require a different theoretical framework and is out of scope.

**Reviewer Scores:**

Reviewer QKDR: Remain 6 as in the reveiwer's reply.

Reviewer R2m9: Likely raise to 4 or 6 as all issues were addressed.

Reviewer fY3w: Likely keep it at 6 or raise to 8  as all issues were addressed.

---

### Decision · Program_Chairs · 2026-01-26

Accept (Poster)